# Doctor Attributes That Patients Desire during Consultation: The Perspectives of Doctors and Patients in Primary Health Care in Botswana

**DOI:** 10.3390/healthcare11060840

**Published:** 2023-03-13

**Authors:** Vincent Setlhare, Sphiwe Madiba

**Affiliations:** 1Department of Family Medicine and Public Health Medicine, School of Medicine, University of Botswana, Private Bag UB, Gaborone 0022, Botswana; setlharev@ub.ac.bw; 2Department of Public Health, School of Health Care Sciences, Sefako Makgatho Health Sciences University, Pretoria 0208, South Africa; 3Faculty of Health Sciences, University of Limpopo, Polokwane 0700, South Africa

**Keywords:** Botswana, consultation, communication skills, doctor attributes, doctor–patient relationship, primary health care, skills

## Abstract

Doctor attributes contribute significantly to the quality of the doctor–patient relationship, consultation, patient satisfaction, and treatment outcomes. However, there is a paucity of research on this topic in many settings in developing countries, including Botswana, where accessibility and availability of care itself are a challenge. The study examined doctor attributes that patients in Botswana desire from the perspectives of doctors and patients in selected public clinics located in four health districts of Botswana. We used a qualitative design and conducted face-to-face interviews with 32 adult patients and 17 doctors selected through the purposive sampling technique. Interviews were audio-recorded and transcribed. Data analysis followed the six steps of qualitative thematic data analysis. We found both discordance and congruence between the doctors and patients on key attributes that patients desire in a doctor during consultation. Both agreed that effective communication and listening skills were key desirable doctor attributes that improve the doctor–patient relationship. Conducting the consultation in the language of the patients enhances effective communication. Doctors cited clinical expertise and competence as key desirable doctor attributes, whereas patients cited interpersonal and social attributes including kindness, empathy, and respect as key doctor attributes that increase trust in the doctor. However, patients expected the doctor to have clinical knowledge, which they perceived as essential to improve doctor–patient interaction and health outcomes. The findings highlight a need to enhance the interpersonal and communication skills of doctors to improve the quality of doctor–patient interactions. To optimise and enhance the consultation, continuing professional development should be adopted as a strategy to improve the communication and interpersonal skills of doctors.

## 1. Introduction

The World Health Organisation (WHO) recognises the relationship between doctors and patients as central to the delivery of good quality health care [1]. The doctor–patient relationship is the receptacle within which the consultation occurs and is crucial to health care delivery [2,3], whereas the consultation is deemed as the foundation upon which patient care is built [4], and serves to generate a common understanding of what the patient hopes to gain from the consultation [5]. The consultation is fundamental to the delivery of primary care and, when done well, it may lead to a good diagnosis from which good treatment of the illness may emanate. Thus, the consultation is regarded as a gateway to and an important determinant of good patient care. However, different ways of organising consultations may lead to different patient experiences [6].

To provide the patient with quality care, the doctor–patient relationship needs to be healthy. It is therefore, the bedrock of medicine through which a doctor and a patient consultation is accomplished [7]. It is regarded as a fundamental component of the care process that might improve health outcomes of patients particularly in primary health care (PHC). While the importance of the relationship between doctor and patient during consultation is widely acknowledged, there is substantive evidence that the relationship is characterised by challenges [8,9]. In the African context, research established that patient satisfaction with physician interaction is generally low [10]. The success of the doctor–patient interaction during consultation depends not only on the doctor’s clinical knowledge and skills, but also on the nature of the relationship that exists between them [3,11]. The doctor–patient relationship provides the primary means for the diagnosis and treatment of disease. Thus, a good doctor–patient relationship and high-quality consultation increase patients’ willingness to reveal information, describe symptoms, adhere to a treatment plan, and achieve desired health outcomes [12,13]. Therefore, it is important for doctors to recognise when the relationship is challenged as well as the factors contributing to the poor interactions, in order to improve care [9].

Patients base their assessment of a consultation on the doctor’s qualities and attributes, which influence the outcome and quality of the consultation [12]. Doctors’ attributes and interaction with patients during consultation affect the health outcome, adherence to treatment, and health-seeking behaviours of patients [14,15]. Doctor attributes are the characteristics or behaviours that doctors display or have [16]. The doctor–patient relationship has evolved over time and the medical profession has adapted to global migration, which has altered doctor–patient interactions [17], as well as medical education and technology [18]. Doctor attributes are described in terms of how a doctor relates to patients, and the character, knowledge, and skills the doctor displays during consultation [16]. Therefore, patients desire a doctor to have good interpersonal qualities such as empathy, compassion, and a caring attitude. In addition, patients desire doctors to be knowledgeable, skilled, competent and have good communication skills [19,20,21,22,23]. In a recent survey with patients and doctors, patients described good doctor attributes as knowledgeable, caring, professional, excellent, and competent [24].

Of note is that there are differences in constructions and perceptions of desirable and undesirable doctor attributes from region to region around the world [3,16,18,25]. This could be attributed to the fact that doctor attributes that influence the doctor–patient relationship are shaped in different cultures, societal norms, and institutions around the world [18,26,27]. Furthermore, doctors’ perspectives of the doctor–patient relationship and doctor attributes may differ from those of patients, and these views influence the perceived quality of care rendered [2]. Doctors are often unaware of whether patients are satisfied with a consultation because patients tend to be polite and show respect during consultations [10]. Research shows that doctors emphasise competency and good clinical skills as the key attributes that patients desire in the doctor [22,23,28], whereas a systematic review of 57 qualitative studies from North America, Europe, and Australia showed that patients liked doctors who were good listeners, empathetic, respectful, kind, and humorous [29]. The importance ascribed to a doctor’s attributes during consultation suggests that more training in non-cognitive skills should be included in medical education [21].

The association of doctors’ attributes and health outcomes, adherence to treatment, and health-seeking behaviours of patients, has been well established [10], particularly in developed countries [14,30]. Desirable doctor attributes displayed during consultation improve the doctor–patient interaction and enhance the trust that patients have in their doctors [10]. However, few studies have investigated the perceptions of a good doctor from the perspective of diverse populations in low and middle income countries (LMICs) [17]. Thus, in many settings in LMICs, including Botswana, there is a paucity of research on this topic, and perceptions about desirable and undesirable doctor attributes are informed by research from developed countries [17]. Moreover, the research on doctor attributes done in developed countries lacks detailed data as to what doctors and patients agree upon concerning desirable attributes.

Investigating the role of doctor attributes in patient healing and health outcomes in LMICs such as Botswana is crucial given the changes in health care delivery systems due to limited health resources [31,32]. Furthermore, an understanding of the qualities that patients desire from the doctor–patient relationship is of relevance in primary health care settings since the first contact of patients with the health care system usually occurs there [33]. Moreover, desirable doctor attributes in and of themselves make doctors agents of healing, independent of the interventions they prescribe [34,35]. It is important that guidelines for a quality doctor–patient relationship during consultation is informed by appropriate and contextual research.

Therefore, the aim of this study was to explore from the perspectives of patients and doctors the doctor attributes that patients in Botswana desire and those that they do not desire and assess how the doctor and the patient should relate. A better understanding of the doctor–patient relationships and how that affects patient care is essential [33]. Therefore, the study findings will provide an understanding of the desirable doctor attributes to inform the consultation process in primary health care facilities, as well as how doctors are trained and health care is organised.

## 2. Materials and Methods

### 2.1. Study Design and Setting

This qualitative study used semi-structured in-depth interviews that were conducted with doctors and patients in primary health care facilities to explore doctor attributes that patients in Botswana desire. We conducted the study between September 2018 and October 2019. Botswana is a landlocked country in southern Africa, which has arid, semi-desert conditions in more than half of the country. The country shares borders with South Africa, Namibia, and Zimbabwe. Botswana has a population of about 2.5 million people, of which more than two thirds of the population live in rural areas [36]. Healthcare in Botswana for 93% of the population is organized based on a publicly funded primary healthcare model. Health districts and local authorities are responsible for the delivery of primary health care (PHC) services through health posts, which serve remote and rural areas, clinics, primary hospitals, and district hospitals. Most primary health facilities are run by nurses who consult and treat patients. Depending on the availability of doctors, some clinics had one or two doctors in attendance daily or at least two days per week. Over 95% of the total population are within 5 km of a healthcare facility.

The study setting was seven public clinics located in four health districts of Botswana. The clinics were selected purposively such that there was a national spread of clinics to cover most regions of the country. The selection of clinics further ensured the geographical, tribal, and socioeconomic diversity of patients attending clinics in Botswana. All selected clinics have a doctor present at least once a week. Clinics provide mother and child services, HIV/AIDS, TB, non-communicable diseases, dental care, and laboratory services.

### 2.2. Study Population

The study population comprised patients aged 18 years and older who presented to doctors with a variety of diseases in the selected clinics. The number of patients ranged from 700 to 1000 per clinic per month. The study population also included doctors who visit clinics at least once a week. The research assistant (interviewer) recruited all the study participants through purposive sampling to select patients and doctors who could relate what they understood to be desirable and undesirable doctor attributes. To recruit doctors, the interviewer informed them about the study during their lunch or team break. Doctors who had practiced medicine in Botswana for two years or more and worked in the selected clinics during the study period were eligible to participate. Appointments to conduct the interviews were scheduled for those who volunteered to participate and meet the eligibility criteria. All interviews were conducted at a time and place convenient to the doctors. None of the interviews were conducted on the same day because of the high patient load in the clinics, but at a later date outside of the health facilities.

To recruit patients, the interviewer was assisted by the clinic staff who introduced the interviewer and the study to patients in the clinic. The interviewer recruited patients on an individual basis while waiting to see a doctor, to receive medication at the pharmacy, to get a dressing, or to give a laboratory specimen. The researcher selected patients who were 18 years and older and had consulted a doctor at least twice in the previous 12 months. The principle of maximum variation was also considered and applied in the sampling to ensure that the sample reflected a diverse group of participants [37]. Employing maximum variation sampling ensured that essential and variable features of the consultation as experienced by diverse patients informed the understanding of the desired doctor attributes [38]. The researcher selected and included patients from different backgrounds such as different age groups, gender, occupation status, educational attainment, and medical conditions.

In qualitative research, data collection is guided by saturation [37], a point in data collection and analysis when new incoming data produces little or no new information to address the research question. In this study, data saturation occurred after 32 patient and 17 doctor participants were interviewed. Data saturation was deemed reached when no new themes were coming out from each set of interviews (patients and doctors).

### 2.3. Data Collection, Tools, and Procedures

Data were collected using semi-structured interview guides with open-ended questions. Two interview guides—one for doctors and one for patients—were designed after reviewing the literature on the topic [15,39]. The English patient interview guide was translated into Setswana by the lead author who is a Motswana person who speaks the language at home and at work. A research assistant (interviewer) who speaks Setswana and has a university degree back translated the interview guide. The patient and doctor interview guides asked four broad questions with probes and follow up questions that included in the guide.

Prior to implementation of the research project, the interviewer who conducted all interviews underwent training on qualitative research, facilitated by the second author. The training focused on a comprehensive understanding of the study protocol and objectives to ensure that the collected data would answer the research questions. In addition, the interviewer was trained in issues of confidentiality, maintaining privacy, administering informed consent, interviewing skills, and the use of probes. Training was followed by a pilot study conducted with patients in one of the selected clinics. The pilot study assessed participants’ understanding of the questions, familiarised the interviewer with the application of the interview guide, and appreciated the length of time required for the interview. No changes were made to the patients’ interview guide.

The interviews with patients were conducted on the day of recruitment in a private room reserved for the interviews in each clinic. All interviews were conducted in Setswana and audio recorded after giving a full explanation of the research, discussing ethical aspects that the participants needed to be aware of, and obtaining written informed consent. The interviews with doctors were conducted on an appointment basis at an agreed upon time that was convenient to the doctors. All interviews were conducted in English in the doctor’s consultation rooms in the clinic or a quiet place away from the clinic, such as a restaurant.

Patients and doctors were interviewed in the same manner. All interviews lasted for 30 to 60 min, and field and interview notes were collected. At the end of the interview, a short demographic tool was administered and captured data pertaining to the patient’s age, gender, level of education, employment status, and location of the patient’s clinic. Concerning the doctors, the tool captured their age, gender, number of years of work experience, number of years of working in Botswana, and the location of the clinic where the doctor was working. Small tokens of appreciation (BWP 30 or some refreshments) were given to participants at the end of the interview. Doctors were given refreshments while most patients were given cash to cover transport fees.

### 2.4. Data Mamangement 

To preserve confidentiality and the safety of data obtained through interviews as well as the protection of study participants, the audio recordings of the interviews and typed transcripts were assigned unique corresponding identifiers. After each interview, the interviewer downloaded the audio recorded file in a password-protected folder in a laptop kept in a lockable cabinet in the investigators office. Only the investigators had access to the transcripts and audio recordings, which will be stored for five years. All interviews were conducted in a private space and pseudonyms were used.

### 2.5. Data Analysis 

The data was analysed for themes, following the steps outlined in the literature [40]. In a thematic analysis, the researcher looks for recurring information or patterns within the data. The first step undertaken was the transcription of audio files verbatim in Setswana and checking the correctness of the Setswana transcripts by listening to the audio tapes while reading the transcripts. Next, the authors independently read the patient and doctor transcripts several times to familiarise themselves with the data. This step was followed by the authors independently analysing four patient and three doctor transcripts line by line to search for meanings and patterns that appear across the transcript and identifying initial emerging codes using manual coding. This process generated 160 codes from four patient data transcripts and 116 codes from the three doctor data transcripts. The authors used the initial codes to develop two codebooks (one for patient data and another for doctor data).

For both patient and doctor data, the authors next engaged in a rigorous process to search for emerging themes and subthemes through the process of merging similar codes. Then, the authors agreed on the definition and naming of themes and subthemes after several sessions.

The authors revisited the emerging themes several times, redefining and regrouping them, and removing themes that do not have enough data until the final doctor and patient codebooks were made. The authors then uploaded all the transcripts into NVivo 12, a qualitative analysis software package, for application of coding. Analysis continued as the authors searched for relationships between the themes. New codes and themes emerged as more transcripts were analysed. The patient data and doctor data were analysed separately but the findings were later merged.

The authors used the emergent themes and subthemes to write the narrative report and present the findings that reflect the desired doctor attributes.

### 2.6. Rigour

To ensure rigour and credibility, we used strategies as recommended in the literature [41]. Prolonged engagement and familiarisation with the data was achieved through listening to the audio recordings of the interviews repeatedly and translating the transcripts into English. We employed investigator triangulation throughout all the stages of data analysis and interpretation to enhance the credibility of the findings and reduce investigator bias [42]. Other strategies that we used to enhance credibility included employing a good quality digital recorder, transcribing the audio recorder verbatim, and using the Nvivo12© software programme for data analysis. It was important that the lead author, as a specialist family physician, performed bracketing throughout the data analysis to reduce inherent biases that might influence the interpretation of the desired doctor attributes [43].

## 3. Results

### 3.1. Sociodemographic Profile of Participants

The 32 adult patients who participated in this study had consulted with a doctor at least twice in the previous 12 months. Most of them were female (22/32) and their ages ranged from 19 to 67 years with a mean age of 38 years. With regards to educational attainment, most (18/32) had secondary or tertiary school education and half (16/32) were employed. These patients were selected from PHC clinics in rural and urban districts of Botswana, and 13/32 lived in a rural setting while the rest lived in a small town or a city (Table 1).

With regards to doctor participants, 17 doctors were interviewed and 10/17 were female. Their ages ranged from 28 to 58 years with a mean age of 37 years. Most of them (10/17) had more than five years of work experience in Botswana. Similar to the patient participants, the doctors were providing health care services in PHC clinics in various urban and rural districts of Botswana (Table 2)

### 3.2. Themes

Three themes emerged from the analysis of patients and doctors’ data on the attributes doctors should possess and display during a consultation in PHC clinics, namely (1) communication skills, (2) good interpersonal skills, and (3) good clinical skills. The themes and eight subthemes are summarised in Figure 1.

#### 3.2.1. Communication Skills

Communication is considered crucial in building a healthy doctor–patient interaction and emerged as an important desired doctor attribute and a key feature of a successful doctor–patient interaction among the patients and the doctors. Doctors believed that having good communication and listening skills is important for the consultation to be successful.

“When I talk, the doctor should be able to understand what I am talking about. They [doctor and patient] should talk to each other well so that both can hear what the other one is talking about”.(27-year-old female)

“So, you would answer the patient’s questions to clear the confusion that they have. That is what they like, and really explaining to them, they mostly want to understand”.(37-year-old doctor with nine years of service)

“A good consultation means open relations, or open talks or no hidden stories or the patient being able to express his/her condition, the doctor being able to understand their condition...”.(31-year-old male doctor with six years of service)

Ability to communicate in local language

Patients desired to be consulted by a doctor who speaks their language. They felt that the doctor–patient interaction was improved if the doctor and patient spoke a language they both understood and reported that being consulted in English was a barrier to communication during the consultation.

“We should speak to each other in a language that I understand, and the doctor understands. If the doctor is a foreigner and speaks English, there should be an interpreter who can translate well. It is important because I will be able to understand many things, and I am able to explain my illness well to the doctor, when I am ill, if we understand each other”.(25-year-old female)

“It is because I want to relax and speak in my own language, you see”.(36-year-old female)

“Patients like it when you explain things to them. They prefer if you speak the language they can speak so they can express themselves”.(29-year-old doctor with four years of service)

“Like I have already said, it is the type of doctor that only speaks English of which all will be lost for you if you do not know English. It is because that particular doctor and patient will not be understanding each other”.(19-year-old female)

Active listening

Both the doctors and patients indicated that patients desire to be listened to, not to be rushed, not to be interrupted, and to be provided with adequate explanations about their condition during the consultations. Effective listening is essential for clinical data gathering as well as fostering the doctor–patient relationship. Patients were appreciative when doctors listened and respected their opinions. This made them share more information with the doctor, which fostered a good doctor–patient interaction.

“I think what patients like is the doctor who is friendly first of all, a doctor who is a good listener a doctor who further investigates their problem, like examine their problem area”.(28-year-old female doctor with three years’ working experience)

“I want a doctor who is patient. A doctor who listens when I speak to him”.(35-year-old female)

“There should be clear understanding, the doctor must listen very well to the patient and in return the patient must also make sure they understand the doctor’s instruction”.(female, 54-years-old)

“So you should listen to the patient, then explain everything; where he doesn’t understand you should be able to explain”.(30-year-old female doctor with five years of service)

Furthermore, the doctors were of the view that allowing the patient to talk uninterrupted encouraged the patient to convey more information about the nature of his or her illness, which is conducive to an effective consultation and enhanced patient care.

“When you talk to a patient, he does not want to be interrupted. While he is still talking, do not just start recording and say done!”.(30-year-old doctor with five years of service)

“Doctors who really show confidence and show understanding of their work will let their patient feel at ease and let their patients narrate their conditions softly and easily. Imagine if you did not hear your patient well. That can lead to wrong prescriptions. Then you can do wrong diagnosis which will lead to giving wrong medication”.(31-year-old doctor with six years of service)

#### 3.2.2. Good Interpersonal Skills

The data revealed certain behaviours, manners, and attitudes of the doctor that are expected during the consultation. The patients value being consulted by a doctor who is friendly, respectful, attentive, patient, interactive, caring, listening, and treated patients as people. The doctor’s good interpersonal skills resulted in patients being satisfied with their consultation and treatment outcomes.

Kindness

The doctors and patients agreed that patients expected to be treated with kindness and compassion by doctors during the consultation. They believe that patients value a doctor who is friendly and caring.

“You see when you get into the consultation room you see a smile. When you greet the doctor, he/she returns your greeting in a loving way. Even during the consultation, you connect with the doctor, and you engage in small talk and laugh together”.(36-year-old female)

“You have to be easy to talk to, be approachable, relatable. You have to be relatable…, you cannot come with a heavy make-up and expect to be approachable you know what I mean?”.(29-year-old female doctor with five years of service)

“The doctor I would like to treat me is a doctor who has botho [kindness]. He should be patient and should not be short tempered with patients”.(31-year-old female)

“A doctor, who is kind, is very good because he is able to help you nicely and explains to you what is happening”.(31-year-old female)

Compassion 

The interviews revealed that doctors understood that they needed to show compassion and empathy because empathy is valuable to the building of good–doctor patient interactions and relationships. Both believed that kindness and compassion create a good environment for the consultation, create trust in the doctors, make patients believe that doctors are empathetic, and improve health outcomes.

“I want a type of doctor like the one who I consulted with today who is well behaved, relaxed, patient, empathises with a patient, and is a good listener who can come up with solutions to problems and make you feel better about your condition”.(19-year-old female)

“Patients are already in a state of distress okay. So compassion just simply means you identify with them and you show understanding that…, you understand what they are going through”.(40-year-old doctor with three years’ practice)

“And then also you need to have compassion. You need to be able to identify with the suffering of the patient as well. Yeah, in addition, also I think patients expect you to show empathy”.(40-year-old doctor with three years’ experience)

Respect

Patients want doctors to be respectful towards them during the consultation and doctors indicated the need to demonstrate respect for the patients. When doctors act respectfully towards patients, the doctor–patient interaction is improved. They mentioned the issue of the doctor talking on the phone, being on Facebook, and using Twitter as being disrespectful to patients.

“I could say that as health professionals, we should be able to show respect to our patients, be open minded, be professional, and show patients that they are most welcome to your consultation room, because that is what will determine the good consultation process”.(31-year-old male doctor with six years of service)

“Well…, good doctors do not get busy with the phone or computer, they listen; they don’t get distracted by Facebook and as patients we appreciate that; because doing that is being disrespectful to us and we can just get up and leave anytime. I don’t want a disrespectful doctor”.(54-year-old female)

“Being respectful in the sense of the words that you use, using polite words, excusing yourself politely you know what I mean…, the way I talk to them with a nurturing voice”.(29-year-old doctor with five years of service)

“I think the doctor and patient have to talk in a way that shows mutual respect”.(40-year-old doctor with three years practice)

“I believe that when the patient and the doctor are in the consultation room, everything should go well, there should be mutual respect and love together with freedom of expression”.(19-year-old female)

#### 3.2.3. Good Clinical Skills

Although clinical knowledge and skill was one of the key doctor attributes that the doctors identified as desired by the patients, they also mentioned that they expect the doctor to be competent and knowledgeable.

Professional competency

The doctors and patients believed that a doctor with good clinical skills would make correct diagnoses to manage patients appropriately. The interviews further established that patients preferred a doctor who is older, experienced, knowledgeable, and demonstrates professional competency.

“You see this issue of the doctor knowledge; it gives the confidence to the patients. That whatever they are getting is the correct treatment. If the doctor doesn’t show knowledge the patient begins not to trust the treatment plan the doctor is going through”.(50-year-old male doctor with fifteen years worth of experience)

“Then I think ultimately you also have to be competent; I think patients expect the doctor to be competent”.(40-year-old doctor with three years’ worth of experience)

“It is a fact that if a patient does not believe that the doctor knows what he is doing he will not believe that the medication you are prescribing them to deal with the condition is going to work”.(50-year-old doctor with fifteen years’ worth of experience)

The interviews further established that patients preferred a doctor who is older and experienced. They associated these attributes with clinical skills. They expressed that an experienced doctor knows when to refer them to a medical specialist for further treatment, when indicated.

“I prefer to be treated by a mature doctor. The older doctors who show that they have been in the field for a long time and through his experiences, he has met a number of conditions of which maybe my condition he has met it before. So…, that helps in me being given the correct treatment which will help me address the issue that is affecting me”.(45-year-old male)

“Like I had said about the help I was given when I got here, as the doctor was helping me, knowing that there are others who know more than him, he saw it better to refer me to them. I went to XXX eye clinic where I got another assistance, which resulted in me getting better. My belief is that the doctor I started with in YYY is the one who really helped me because if it were not for him, I would not have even gone to XXX because I did not know where to go for a specialized eye doctor. I only know our clinic”.(48-year-old female)

Performing thorough examinations

Performing thorough examinations was identified as another attribute that was related to good clinical skills by both doctors and patients. Doctors believed that patients wanted to be examined by a doctor when they consult. Performing a thorough examination is important for the doctor to arrive at a good diagnosis, and to prescribe the appropriate treatment.

“During consultation, the doctor should not just write down what they are saying without doing examination. Do not cut corners…, if you don’t do examination, they will think you were not thorough”.(30-year-old doctor with five years of service)

Allocating enough time for the consultation

Both the doctors and patients mentioned that allocating enough time for the consultation is desirable. They stated that spending enough time with a patient improves the chances of correct diagnosis and care since doctors require adequate time to perform a thorough examination to come to a correct diagnosis. Allocating adequate time for the patient to explain their symptoms and condition is critical for a quality consultation and building trust that the doctor would come up with a good treatment plan.

“The doctor must have time for the patient to explain everything that the patient wants to…, what the patient needs to know”.(40-year-old doctor with three years practice)

“You should give the patient time to talk, even though from time to time you would interrupt asking those open ended questions”.(30-year-old doctor with five years of service)

“So, if you take your time, you hear everything when you get a history, and if you get a full history, you can treat better, and everything comes out if you take your time. When you rush not everything is said and some things that are important can be missed. I take my time honestly”.(29-year-old doctor with four years of service)

“I became happy that this doctor is helpful; he was taking a lot of time, but I understood that that was because he was doing it with love”.(26-year-old female)

## 4. Discussion

The study explored the doctor attributes that patients in Botswana desire with 17 doctors and 32 adult patients selected from PHC clinics in rural and urban districts of Botswana. We found that while doctors identified clinical expertise and competence as key doctor attributes, patients reported that interpersonal, social, and communication skills were more important and desirable than clinical attributes during consultation. The findings in the current study are consistent with other research findings that documented the higher importance that patients place on the doctor’s kindness and empathy, more so than their competence during interactions [11,22,23,28]. Furthermore, there is empirical evidence that quality communication, interpersonal skills, and shared decision making are the most important aspects of high-quality health care that patients across different settings and socio-demographics value [11,21,22,23].

Although patients and doctors differed in what they believed were key desired doctor attributes, there was consensus on other desired doctor attributes. They identified the ability to communicate effectively as a key attribute that patients desire from the doctor. Effective doctor patient communication occupies centre stage in medicine. The doctors and patients concurred that good communication skills promote a mutual understanding of diagnoses and treatment plans. Similar findings that effective communication facilitates a common understanding of the patient’s problem were previously reported in developed countries [44,45]. Consistent with previous findings, exchange of information, sharing ideas about diagnoses and treatments, discussing social issues, and explaining follow up care were identified as critical elements of effective communication [46,47]. The data further revealed that effective doctor–patient communication improves the doctor–patient relationship, thus corroborating the findings of a review of doctor–patient interaction [29,44]. According to the current study, as with those of others, the emphasis is always on the doctor to facilitate efficient communication [33,48].

In addition, the study participants agreed that patients wanted to be listened to and to be given a chance to express themselves fully. Allowing the patient to talk uninterrupted encourages the patient to convey more information about the nature of his or her illness and is conducive to an effective consultation. Consistent with previous findings, the participants stated that effective listening is essential for clinical data gathering, for fostering the doctor–patient relationship, and enhancing quality patient care. In addition, effective listening makes patients feel that they are recognised as human beings and that what they have to say is important [49,50,51].

The current study further revealed the importance that patients placed on the use of a local language during the doctor–patient interaction. Patients in the current study preferred that doctors communicate with them in their own language because it affords them the space to feel comfortable to share personal information, to be understood, to participate meaningfully in their care plan, and to feel respected and accepted. The importance of language concordance with the consulting doctor has been shown in studies conducted in developed countries [52,53,54]. Research has established that language concordance also improves adherence to treatment and the clinic schedule, which improve health outcomes [54,55].

The current study also revealed patients’ preference of being treated by Batswana doctors, similar to findings of a study in Australia, and a systematic literature review of studies that was mostly from developed countries, which showed that patients preferred doctors of the same ethnic group. Ethnic congruence has been shown to be beneficial for mutual understanding between doctors and patients [29,56], The doctor as a person functions as a drug or as a healing intervention long before he has applied medication or performed surgery [34,35]. This suggests, therefore, that a doctor should have inherent personal qualities that make him a healing instrument. This explains why patients and doctors in the current study identified many social skills that were grouped together under the themes of respect and kindness as desirable doctor attributes.

We found that patients needed to be treated with kindness and compassion, which creates trust in the doctors and improves health outcomes. They wanted doctors to be patient, understanding, empathetic, respectful, friendly, and courteous. The WHO’s framework to strengthen the doctor–patient relationship identified the need for dignity, respect, and comfort from the doctor as central to the healing process [1]. The desirable doctor attributes that were derived from doctor and patient data are similar to the findings in other studies in developed countries and in other parts of the world [19,29,46].

Patients desire their doctors to have good consultation and clinical skills to make correct diagnoses. It should be noted that patients associated good clinical skills with being an older, experienced, knowledgeable, and competent doctor. They indicated that they preferred to be consulted by this kind of doctor who would be able to provide a good diagnosis and treatment plan. Accordingly, the WHO framework states that a doctor must be competent and possess sound clinical skills to positively influence the doctor–patient interaction [1]. The findings are consistent with those of studies conducted previously [39], which reported that patients prefer their doctor to have good clinical skills, which improves doctor–patient interactions [13]. Clinical skills and knowledge inspire trust in the doctor and when patients trust their doctors they adhere to their treatment, their health improves, their health seeking behaviours are enhanced, and adherence to clinic follow up visits improves.

The current study showed that patients wanted doctors to allocate adequate time for the examination, take a detailed history, examine them thoroughly, and come up with a correct diagnosis and treatment plan. Similarly, in a study conducted among patients in a family medicine clinic in Nigeria, patients liked it when doctors discussed the cause of the illness with them and took a detailed history [39]. The findings in the current study corroborate those of studies conducted in other parts of the world [21]. The findings established that a thorough examination by a doctor increases patients’ confidence that the doctor will reach the correct diagnosis and appropriate treatment plan.

## 5. Limitations of the Study

This study was conducted among doctors and patients and has the potential for social desirability in the case of medical doctors who may have overemphasized the attributes that patients desired. In addition, selection bias could have been introduced in the study because we selected only patients present at the clinics during the day of data collection. However, this was mitigated by employing maximum variation and sampling of a diverse group of participants in terms of age, gender, occupation status, educational attainment, and medical conditions. With regards to doctor participants, we selected only those who were visiting the clinics during data collection. Doctors who were no longer visiting the selected clinics were excluded, and their views were not captured.

## 6. Conclusions

The study findings highlighted key doctor attributes patients desire during consultation in Botswana. To our knowledge, this is the first study to explore the perspectives of doctors and patients on doctor attributes. The findings suggest that thus far, doctor–patient interaction and consultation in Botswana has been informed by data from western countries without acknowledging the unique context of Botswana.

We found that there was both discordance and congruence between the doctors and patients on the attributes that patients desire in a doctor during consultation. Both the doctors and patients agreed that effective communication and effective listening skills were desirable doctor attributes that improve the doctor–patient interaction, the consultation, adherence to treatment plan, and overall health outcomes.

The findings highlight a need to develop the interpersonal and communication skills of doctors to improve the quality of doctor–patient interactions in public clinics. These could be achieved through continuing professional development programmes that would take into consideration the culture, beliefs systems, and context of the Batswana people. Thus, the findings should inform strategies to optimise and enhance the consultation as one of the primary objectives of the health care system in Botswana.

The study identified several desired and undesired doctor attributes using qualitative research with a limited sample of doctors and patients. There is need for quantitative research to determine the extent to which patients and doctors identify with the doctor attributes identified in this study using larger sample sizes. Results from these kinds of studies could then inform interventions and strategies to enhance the doctor–patient interactions that are influenced by doctor attributes.

## Figures and Tables

**Figure 1 healthcare-11-00840-f001:**
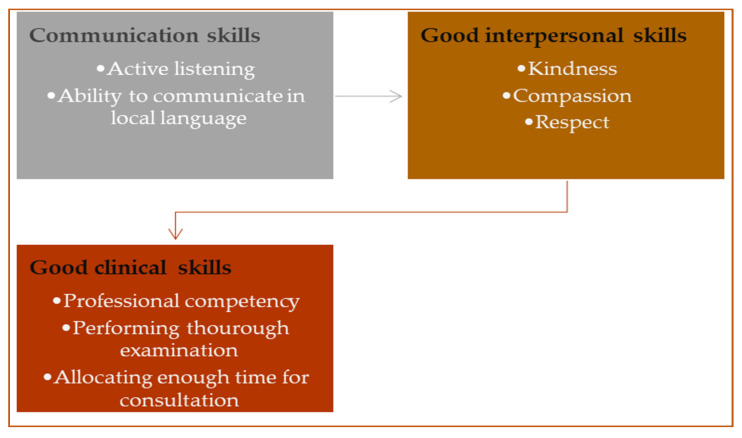
Summary of themes and subthemes.

**Table 1 healthcare-11-00840-t001:** Characteristics of patients attending selected clinics.

Theme	Category	Frequency	Percentage
Gender	Male	10	31.2
	Female	22	68.8
Age group	19–30 years	10	31.2
	31–40 years	10	31.2
	41–50 years	9	28.1
	51–60 years	3	9.4
	61–67 years	2	6.2
Level of education	None	2	6.2
	Primary Education	6	18.8
	Junior Secondary School	9	28.1
	Senior Secondary School	9	28.1
	Tertiary	6	18.8
Employment status	Employed	14	43.8
	Unemployed	18	56.2
Health District	Gaborone	10	31.2
	Maun	9	28.1
	Serowe	8	25.1
	Takatokwane	5	15.6

**Table 2 healthcare-11-00840-t002:** Demographic characteristics of doctors.

Theme	Category	Frequency	Percentage
Gender	Male	10	58.8
	Female	7	41.1
Age group	28–30 years	5	29.4
	31–40 years	7	41.1
	41–50	4	23.5
	51–60	1	5.9
Health District	Gaborone	8	47
	Maun	5	29.4
	Serowe	3	17.6
	Takatowane	1	5.9
Service years	3–5 years	7	41.1
	6–10 years	6	35.2
	11–15 years	4	23.5

## Data Availability

The data that support the findings of this study are available from the corresponding author, upon reasonable request.

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
