# Peer review of "Doctor Attributes That Patients Desire during Consultation: The Perspectives of Doctors and Patients in Primary Health Care in Botswana"

_healthcare, 2023, doi:10.3390/healthcare11060840_

Round 1
Reviewer 1 Report
Thank you for your interesting research. You put a lot of effort into the methodology and therefore know the data very well. It would be interesting for the readers to see a comparison between the two groups surveyed (see “results”). I have made suggestions for better readability and clarification of some open points.
Abstract
The abstract is not divided into introduction, methods, results, conclusion.
Keywords: I would add "communication skills", which is more important for search strategies than e.g. "patients".
Introduction
The introduction is comprehensive and comprehensible. Are there studies from Botswana or comparable countries?
Methods
Please include a graphic or flowchart of how the categories were formed.
2.1 this is not the study desing, but the setting and the socio-cultural/geographical environment.
Information on data protection, data management is missing.
Setting: The reader would like to know how a typical primary care clinic in Botswana is organized: are by nurses?
Please explain the purporsive sampling technique more in detail. In the abstract you mentioned “6 steps”, which are not clearly visible in the text.
How and by whom were the patients selected? How were the doctors selected?
How many questions did the interview guide have? Was it piloted? Who conducted the interviews? Did the patients know beforehand that they would receive money? If yes: please mention this as a limitation in the discussion.
P 4 line 190: were only or 4 interviews coded? Below and in the abstract you mentioned 32 and 17.
Results
A first table “participants” should contain your data of respondents, see other publications. You could also show a little map of Botswana with the locations of the clinics involved in the study.
It is better to give fewer verbatim examples, better show them in a table, which is easier to read. The table would also allow the quotes from patients and doctors to be visible to the same or different codes.
Either italicise all or none of the quotes.
P 5 line 2016: How many patients were approached, how many agreed? The same for the doctors.
Were there findings e.g. on gender and age of respondents?
Discussion
P 9 line 420: Is this assessment clear from the interviews? Were codes counted, scored or weighted? If this is a main outcome, it must also be reported in the results section. If not, all results are equally important, which is totally fine with this study design.
Limitations: please add the selection bias and the fact, that only 2 researches did the qualitative analysis.
Conclusion
You can cut here, there are redundancies for discussion. The reader is interested in: what has the study added? What need for further research arises from this?
Literature
Good and up-to-date. Recent citations from 2022. No comma between the authors' initials if there are 2 first names.
Author Response
Thanks for the valuable comments, herewith find detailed response highlighting how we have addressed comments to the best of our ability. All revised sections in the manuscript are in blue font for ease of reference.

Reviewer 2 Report
The paperpresented is interesting, clear and well structured. The work is well framed in qualitative studies and its methodological approach is in line with the requirements of the scientific community. In this sense, I think the authors have done a good job presenting the framework of analysis, the results, the discussion and the conclusions. It is also important to note that the limitations of the study are tight, although too concise. I think it would be good to expand this section a bit.
The main problem I think is in the methodology. It is not a big problem, but an adjustment should be made to the article. The change I propose is really small, which indicates, once again, the interest of the paper. The change is that section 3.1 becomes part of section 2, since it does not talk about the results but about the sample.
I would have liked the authors to study more social research on the doctor-patient relationship. There is a multitude of works on this subject. However, I understand that the different scientific disciplines have their own approaches, so I do not suggest any changes. In any case, I encourage the authors to read papers in this regard.
Author Response
Thanks for the valuable comments, all revised sections in the manuscript are in blue font for ease of reference.

Round 2
Reviewer 1 Report
The paper has become more readable. Comment No 3: a small graphic showing the main themes and sub-themes would have been helpful.
Author Response
We thank the reviewer for the positive comments on our revised manuscript. We attach a graphic presentation of themes and subthemes
